# Effect of Various Binders on the Properties of Microalgae-Enriched Urea Granules

**DOI:** 10.3390/plants11233362

**Published:** 2022-12-03

**Authors:** Austėja Mikolaitienė, Rasa Šlinkšienė

**Affiliations:** Department of Physical and Inorganic Chemistry, Faculty of Chemical Technology, Kaunas University of Technology, 50254 Kaunas, Lithuania

**Keywords:** *Chlorella* sp., biofertilizers, urea crystals, nitrogen losses, soil protection

## Abstract

As the human population grows and the demand for food grows with it, the recycling, or containment of materials is important for resource consumption. Nitrogen is one of the main plant nutrients, most commonly used as the chemical substance urea. Because urea is very soluble and at a relatively low temperature (50–60 °C) it hydrolyses easily (releases N_2_ and CO_2_) in soil solutions; this is why very large amounts of nitrogen are lost and greenhouse gases are released and this causes serious environmental problems. Therefore, the aim of this study was to create microalgae-enriched nitrogen fertilizers with different binders that inhibit nitrogen leaching from the soil. Binders such as water (W), polyvinyl acetate dispersion (PVAD), molasses (M), potato starch (S), and carboxymethyl cellulose (CMC) were used in this study and their influence on leaching was analysed. Granular fertilizers were produced in a drum granulator and dryer under equal conditions: granulation time was 7 min, granulation took place at a temperature of 50–60 °C, at a drum rotation speed of 26 rpm, with a 5° inclination angle of the drum. The results show that the highest quantity of the marketable fraction was 43.01 (±3.068%) and it was obtained using urea, with 10% (*w*/*w*) microalgae additive, and 11.4% (*w*/*w*) of 5% concentration molasses solution. The granules of the fertilizer marketable fraction are similar in size because the size guide number (SGN) of the granules vary in a narrow range and fall within the interval of 287 to 304; this means that the average particle size is ~3 mm. When different binders were used, the average static crushing strength of the granulated fertilizers was lower (approximately 6–12 MPa) than using water alone (approximately 12–16 MPa), but the lower values still fell into the required range. Additives of PVAD solutions and molasses solutions have been found to retain nitrogen in sand. The method of one-way analysis of variance (ANOVA) was used to evaluate the results.

## 1. Introduction

The growing population around the world directly increases the need for agricultural products. As a result, many more raw materials, fertilizers, other chemicals, etc., are used.

Agricultural production in intensively managed systems in Europe has increased over the last 50 years partly due to the increased use of concentrated mineral fertilizers. On one hand, the very intensive use of soil and various chemicals (fertilizers, pesticides, herbicides) has caused serious soil problems [1,2]. Because most mineral fertilizers used are very soluble, they wash away from the soil easily, and plants do not assimilate them. This process causes environmental problems and 30–40% of nitrogen fertilizers are lost by nitrogen volatilization (NH_4_^+^ -N) and leaching (NO_3_^−^ -N) [3]. The biggest losses occur from easily permeable sandy soils with a low water holding capacity and restricted rooting depths. Globally, only about 10% of land areas are suitable for agriculture [4,5]. The excessive and unbalanced use of concentrated soluble nitrogen fertilizers (usually urea) and consequent nitrogen leaching pose certain environmental problems to the quality of land and groundwater and promote eutrophication. According to the International Fertilizer Association (IFA), statistical data from 2020 showed that urea production capacity was 83,487,000 tons, ammonium nitrogen (AN) 31,215,000 tons, and other solid nitrogen fertilizers such as ammonium sulphate (AS), ammonium calcium nitrate (CAN) were ten times less than urea. The IFA predicts that urea capacity will increase to 107,461,000 tons by 2026 [6]. In some countries, nitrogen leaching is a prime source of water pollution [7,8]. Nitrogen exists in both inorganic (primarily as ammonium [NH_4_^+^] and nitrate [NO_3_^−^]) and organic forms within the soil. Yet, it has long been recognized that organic compounds can make up to 95% of the N in some soils. The amount of mineral nitrogen in soil depends on the soil properties, temperature, precipitation, fertilization, and type of agro-technology used, which varies over the years [9,10]. Scientific studies show that the effects of chemical fertilizers on soil cannot be seen immediately, because the soil, due to its composition, is a strong buffer. However, over time, the intensive use of mineral fertilizers decreases soil fertility and causes an imbalance in various elements [11].

There is no doubt that in order to grow a sufficient yield of plants, concentrate mineral fertilizers are the most commonly used. Therefore, it is necessary to take measures to improve soil fertility and productivity, while reducing the negative fertilizers effects on agroecosystems [2].

There are some simple solutions for improving soil quality. One of them is to establish an optimal ratio of water and nitrogen, or water-saving irrigation, which can reduce NO_3_^−^ leaching [12]. Another option to solve this problem is to apply slow-release fertilizers or increase adsorption sites. The water solubility of nitrogenous compounds could be reduced using physical methods such as the coating or encapsulation of water-soluble materials with outer layers of organic or inorganic materials, and chemical means such as nitrogen conversion to polymeric forms, which have reduced water solubility. Physical methods include a coating of fertilizers with additives: inorganics such as sulphur, clay, phosphates, oxides, or gypsum; synthetic organic polymers such as alkyd, thermoset, or thermoplastic resins; natural organic polymers such as latex and lignin; mixed sulphur-polymer, chitosan [13,14]. Generally, each fertilizer granule is encapsulated or coated with a water-insoluble or water-impervious material. Therefore, water must enter the fertilizer granule through the film and solubilize the plants’ nutrients. Quality-coated fertilizers release nutrients over very long periods and with predetermined release profiles [4,13,15]. However, this is a rather complicated way to solve the problem.

It is well known that in the pharmaceutical, food, fertilizer, or other industrial fields, in the production of a granular product (especially by wet granulation), the effect of the binder is very important. The properties of the final product (granules, pellets, tablets, and sticks), such as compressive strength, granulometric composition, etc., partly depend on the type and amount of binder [16,17,18]. In the case when granulation uses materials characterized by high solubility, light agglomeration, strong adhesion, and cohesion forces, only water or water vapour can be used to form crystallization bridges in granules [19]. However, fertilizers and the raw materials of drugs, especially those of organic origin, are often difficult to granulate, so it is necessary to use additional binders [20,21,22].

One other option is the use of organic fertilizers or bio-fertilizers, which enrich the soil with nutrients and do not reduce soil quality. Organic fertilizers from algae are considered a potential alternative to mainstream synthetic fertilizers as they are rich in macronutrients (N, P, K, Na, Mg, Ca), micronutrients (Mo, Mn, B, Co, Fe, Zn), proteins, carbohydrates, and vitamins, all of which directly help in improving the growth and yield of crop plants. Different scientists [23,24] have presented different compositions of microalgae and the composition of the microalgae used in this work was analysed by determining the concentrations of the main plant nutrients (0.8% N, 10.5% P_2_O_5_, 0.5% K_2_O, 0.32% Mg, 22.00% Ca, and 2.31% Na), microelements (22.8% Mn, 0.01% Co, 0.28% Fe, and 0.23% Zn), and heavy metals (>0.01% Pb and Cd). These results were published earlier [25].

When microalgae are cultivated for commercial purposes, wastewater from treatment plants can be used to get a dual effect: water treatment and biomass production. Furthermore, the ability of algae to fixate carbon dioxide during the growth period can be used to reduce pollutant concentrations in atmospheric air and prevent global warming. Therefore, it is likely that the aim of this work to create new microalgae-enriched fertilizers with lower solubility is highly attractive from the point of view of the environment and bio-economics and can have a positive influence on the sustainability of agroecosystems.

## 2. Results

### 2.1. Granulation of Microalgae-Enriched Urea Granules

The granulation of crystalline urea was performed with the addition of 10% dry microalgae powder to different quantities of five different binders: water (W), polyvinyl acetate dispersion (PVAD), molasses (M), starch (S), and carboxymethyl cellulose (CMC). During the process of granulation of algae-enriched urea, water (amount 12.6–13.9%) was used to objectively assess the influence of other binders. All other granulation conditions (granulation equipment, granulation time, granulation temperature, and amounts of raw material mixtures) were the same. Aqueous solutions of different substances (M, S, CMC, and PVAD) with concentrations ranging from 1.0% to 10.0% and consumption from 11.0% to 11.4% (depending on the substance) were used as the binders for crystalline urea. After granulation, the granules were dried in a laboratory drying oven at a temperature of approximately 60 °C degrees until the granule humidity met the requirements for fertilizers, i.e., less than 2%.

The application, dosing, distribution, and assimilation of granular fertilizers are impacted by the physical properties (particle size, bulk density, particle shape, crushing strength, and others) of the product. The accuracy of product properties is very important to achieve maximum yields at a minimum cost, thus ensuring farm profitability.

### 2.2. Particle Size Distribution of Microalgae-Enriched Urea Granules

The particle size distribution of the granules produced with water and different binders is shown in Figure 1, and the quantity of the marketable fraction (2.0–4.0 mm) can be seen in Table 1.

As can be seen in Figure 1, columns of particle size distribution in general are similar when aqueous solutions of different binders are used. In all cases, the fractions of three different sizes (1.0–2.0, 2–3.15, and >5.0 mm) predominate. However, from the standard deviation presented in Figure 1, it can be seen that the data vary in different intervals and their statistical reliability depends on both the binder and the size of the fraction. For example, when using water or carboxymethyl cellulose (Figure 1a,e), the data dispersion is significantly higher than when using other binders. The number of granules of the larger fraction (>5 mm) varies in a particularly wide range when produced with all binders except starch. The amount of the marketable fraction varies more significantly only when using water and cellulose (Figure 1a,e), so in other cases, the results are sufficiently statistically reliable. Using water and starch solution (Figure 1a,d) resulted in a slightly larger amount (%) of larger granules, while the use of other binders (Figure 1b,c,e) produced a larger amount (%) of smaller granules. When analyzing the particle size distribution, it should be noted that in all cases, a relatively small amount (%) of granules (diameter 2.0–4.0 mm) attributable to the marketable fraction was formed.

The data in Table 1 show that the amount of marketable fraction depends not only on the nature of the binder but also on its concentration or the amount of water used. The granulation of the urea and microalgae mixture with water revealed that the amount of marketable fraction decreases by almost 17.0% with a water quantity increase of 1.3%. Consequently, the quantity of water has a very significant effect on the granulometry of the product. The data show that increasing the amount of water by 1.3% (from 12.6 to 13.9%) reduces the marketable fraction by almost half, i.e., from 38.72 (±1.113) to 21.89 (±2.575%). The concentration of S and CMC solutions also has a similar effect on the granulometry of the product. When the concentration of S and CMC increases by 4%, the amount of the marketable fraction decreases by approximately 5.5% and 13.6%, respectively. The concentration of PVAD solution does not have such a significant effect on the marketable fraction of the product. The concentration of M solution also has an insignificant influence on the granulometry of the product, because when the concentration changes from 2.5% to 10.0%, the amount of the marketable fraction changes by 6.10%. However, it should be mentioned that this is not a linear correlation, because, at 5.0% M solution concentration, a maximum marketable fraction of 43.01% (±3.068%) was obtained. The highest marketable fraction was obtained in each series of binders: 38.72% (±1.113%) with 12.6% water; 36.18% (±2.787%) with 5% PVAD solution, 43.01% (±3.068%) with 5% molasses solution, 32.34% (±1.451) with 1% starch solution, and 40.25% (±2.150%) with 1% cellulose solution. Among the five binders used, the highest marketable fraction 43.00% (±3.068%) was obtained using an 11.4% (*w*/*w*) quantity of 5% concentration M solution.

Compared with industrial urea used for fertilization, the granules of this granular algae-enriched nitrogen fertilizer are larger, and the amount of marketable fraction is less because in industrial urea, the marketable fraction is 1.7–2.34 mm and it accounts for up to 99% [26]. However, these differences are fully understood and explained by different chemical compositions (nitrogen only or nitrogen, algae, and binder), different raw material phase states (melt or wetted powder), and production methods (granulation tower or drum granulator).

In order to evaluate the influence of different binders on the granule’s size and structure, optical photographs of the surface of the granules were taken. In Figure 2a,c,d, white binders (PVAD, S, and CMC) can be seen, and Figure 2b shows the brown binder molasses.

### 2.3. Static Crushing Strength of Microalgae-Enriched Urea Granules

Another very important property that characterizes granulated fertilizer particles is the static crushing strength of the granules, which differs greatly depending on the chemical composition. The average values of the static crushing strength of the granules obtained using different binders are presented in Figure 3 and are marked with different color lines for different binders. In order to provide more detailed data, Figure 3 shows the limit values (maximum and minimum) of the granule static crushing strength. Analysing crushing strength measurement data of 2.0–3.15 mm and 3.15–4.0 mm granules (both are classified as a marketable fraction), it can be stated that the average strength values varied over a large range, i.e., from 5.87 MPa to 15.82 MPa and from 7.25 MPa to 13.87 MPa, respectively. Also, it can be seen that the crushing strength depends on the nature of the binder and its concentration, but there is not a linear correlation between those parameters. The average strength values of the marketable fraction of granules (2.0–4.0 mm) produced using PVAD, M, or S solutions are similar, and range from 6.12 MPa to 12.02 MPa, from 7.75 MPa to 11.70 MPa and from 6.31 MPa to 12.70 MPa, respectively. The average crushing strength values of the granules in which the CMC solutions were used as a binder are slightly lower than others and range from 5.87 MPa to 9.44 MPa. The highest average crushing strength values (11.61–15.82 MPa) of 2.0–4.0 mm size granules were obtained when they were granulated using a different amount (12.6–13.9%) of water in the raw material mixture. The nature and concentration of the binder play roles in developing the strength of granules. However, because the main raw materials were the same, the influence of different binders in many cases was not very large or significant. It can be assumed from the results presented in Figure 3 that the crushing strength values of fertilizer particles n with a diameter of 2.0–3.15 mm are insignificantly higher than those of particles with a diameter of 3.15–4.0 mm.

Analyzing the statistical granule crushing strength data (Figure 3), it can be seen that especially large dispersions are obtained when water was used as the binder. For example, in a mixture of raw materials with 13.3% water, the average value of the static strength was 15.82 MPa (2.0–3.15 mm) and 12.06 MPa (3.15–4.0 mm), and the interval between the minimum and maximum values was 7.65–35.56 MPa and 4.44–24.33 MPa, respectively. In other cases, the differences between the minimum and maximum values of the static crushing strength determined in the sample of the same type of fertilizer were smaller but sufficiently significant. Therefore, it can be said that crushing strength data are not statistically very reliable.

### 2.4. Hygroscopicity of Microalgae-Enriched Urea Granules

Water and saturated sodium nitrite (NaNO_2_) solution were used in desiccators to determine the hygroscopicity of the microalgae-enriched urea fertilizer (Figure 4). During the analysis, the temperature in the desiccator with a water vapour environment ranged from 20.2 °C to 21.1 °C (Figure 4a) and the humidity ranged from 99% to 100%. In the desiccator filled with saturated NaNO_2_ solution, the temperature was 20.6–21.2 °C, and the humidity was between 65% and 66% (Figure 4b). The curves in Figure 4a show that all polynomials, except for the one produced with PVAD solution (5% and 7%) are rising evenly and water absorption takes place because the water vapour pressure of the air exceeds the water vapour pressure of the fertilizer. This means that only samples of fertilizers with higher concentrations of PVAD binder are not hygroscopic in these critical conditions. In other cases, in the water environment, the mass of fertilizer increased by more than 300% in 12 days and was still rising.

The polynomials of Figure 4b show that fertilizer samples are of low hygroscopicity when stored in a saturated NaNO_2_ solution environment where the water vapour pressure is much lower. The maximum change in mass was 7.4% when the fertilizer samples were granulated using a 2.5% M solution. Some samples (e.g., with 1% CMC) when stored in a desiccator became over saturated and NaNO_2_ did not absorb but desorbed moisture, resulting in a negative change in mass. It can be stated that the microalgae-enriched urea fertilizer is not hygroscopic under ordinary conditions of fertilizer handling and storage and does not, therefore, require a special coating.

### 2.5. Other Properties of Microalgae-Enriched Urea Granules

#### 2.5.1. pH

As the results (Table 2) show, the pH values of the 10% solution of the microalgae-enriched nitrogen fertilizer depended little on the nature of the binders and their concentrations and ranged from pH 5.85 to 6.65, and the pH value of the 10% aqueous solution of urea was 5.55 (measured). This meant that the addition of microalgae and binders makes the fertilizer less acidic. It can be argued that when molasses solutions are used as a binder, the fertilizers are least acidic (pH 6.40–6.65) because M solutions are weakly alkaline (pH ranges in interval 7–8) and the pH of 10% microalgae solution is neutral (measured pH value 7.0). In the determined interval, the most acidic fertilizers were produced with the use of water and PVAD solutions (pH 5.85–6.10), however, in all cases fertilizer pH values are acceptable.

#### 2.5.2. Moisture Content

The obtained results show that the moisture content of the granules varied within the range of 1.01% to 1.41%, which is more than the moisture content of industrially coated urea (0.3%). However, the obtained result fully complies with the requirements for bulk fertilizers. The wettest granules of the granulated product (moisture content 1.29–1.41%) were obtained when molasses and potato starch solutions were used as the binder.

#### 2.5.3. Bulk Density

The bulk densities (loose and tapped) of a fertilizer provide information relative to the required size of packaging materials, storage houses, stock rooms, etc. Bulk density differs between fertilizer types and is very much related to particle size distribution and segregation. Because bulk density correlates with particle density and has a direct impact on the spread width of fertilizer on the soil, a loose bulk density of 2.0–3.15 mm and 3.15–4.0 mm was measured. The bulk density of 2.0–3.15 mm granules varied within the range of 305.1–430.0 kg/m^3^, and the 3.15–4.0 mm granules range from 331.8 kg/m^3^ to 418.1 kg/m^3^. Summarizing, it is necessary to state that these microalgae-enriched urea granules have a low bulk density and, therefore, their container cannot contain a large amount of fertilizer, and also these fertilizers cannot be spread very widely. However, since the bulk density values are similar, fertilizer segregation during transportation can be prevented.

#### 2.5.4. Size Guide Number (SGN)

It can be stated that granules of the marketable fraction of produced fertilizers are similar in size because the SGN values determined in this research vary in a narrow range and fall within the interval from 287–304 (Table 2); this means that the average particle size is ~3 mm.

### 2.6. Nitrogen Leaching from Microalgae-Enriched Granular Urea

As can be seen in the diagram (Figure 5), nitrogen was already found in the 3rd stage in only a few samples. In parallel with the leaching experiment for all kinds of fertilizer samples (with all binders), the investigation into pure urea leaching was also performed. The results indicated that the addition of 10% microalgae and a different binder strongly affected the behaviour of urea solubility.

As the data in Figure 5a shows, the total concentration of nitrogen in the solution after filtration through sand can be reduced from 33.14% from pure urea to 20–26% from microalgae-enriched urea. This means that loose nitrogen was reduced by 22–40%. It should be noted that most of the nitrogen was leached out the 1st time, but the use of microalgae lowers the extent of leaching compared to that of pure urea. When analysing the influence of different binders on nitrogen leaching, it can be seen that the addition of PVAD (Figure 5b) slowed the urea solubility the most, because in the 2nd stage of leaching a sufficiently high concentration of nitrogen was found in the filtrate. The addition of molasses also had a fairly good effect, as using a 10% M solution as a binder (Figure 5c) resulted in nitrogen being detected in the filtrate of the 3rd leaching stage. The addition of starch and cellulose was less effective in slowing down the dissolution of urea.

PVAD and M (specifically a 10% concentration) solution slowed down the solubility of microalgae-enriched urea fertilizers and reduced the concentration of nitrogen leaching from the sand, and at the same time, this affects underground water pollution. However, nitrogen losses due to emissions were not avoided.

In summary, it can be stated that microalgae (10% *w*/*w*)-enriched nitrogen fertilizers with an average nitrogen concentration of 41.4% (N concentration various 39.9–42.9), created and produced in the laboratory, in many cases have better properties than urea (higher static crushing strength of granules, higher pH values, lower hygroscopicity, dissolves more slowly so less loss). However, a direct comparison is not possible, because in industry, pure urea is produced from melt-in prill towers, and in order to produce these microalgae-enriched nitrogen fertilizers, it is necessary to use a drum or plate granulator.

## 3. Discussion

The particle size distribution is important for the spreading properties and segregation tendencies [27]. Meanwhile, for fertilizer manufacturers, one of the most important properties is the quantity of the marketable fraction in the granular product.

Our analyzed data are presented as optical photo fragments with different binders when the maximum marketable fraction was obtained with each of them. Instead of typical needle-like morphology urea crystals, co-crystals, which are tightly bound by different binders, can be seen in the pictures. This means that during granulation, urea interacted with aqueous solutions of various binders, and urea recrystallized. These data reasonably coincide with observations obtained by other researchers [28,29]. The discussed data allow us to say that the use of a suitable binder is key to the production of a suitable size granule, and this is corroborated by other studies [30,31] which maintain that the type of binding material was the most crucial factor in the granulation process. Authors should discuss the results and how they can be interpreted from the perspective of previous studies and of the working hypotheses. The findings and their implications should be discussed in the broadest context possible. Future research directions may also be highlighted.

In summary, in all cases, static crushing strength values of microalgae-enriched urea granules are higher than 2 MPa (2 MPa or 20 kg/cm^2^ which is an acceptable limit for the crushing strength of fertilizer granules) and in many cases, higher than the crushing strength of industrial urea granules 7.8–8.2 MPa; therefore, there is no risk of the fertilizer particles being broken down during transportation or fertilization [32].

In addition to the granulometric distribution and static strength of the granules, hygroscopicity is a very important property of bulk fertilizers. All fertilizers are more or less hygroscopic, which means that they start absorbing moisture at a specific humidity level or certain water vapour pressure. The absorption of moisture during storage and handling reduces the physical quality [27]. Changing the air temperature and humidity can help to determine conditions under which water absorption will take place. Pure urea does not tend to absorb moisture; when exposed to air under ordinary conditions, it becomes a remarkably hygroscopic substance if the humidity of the air is at a relatively high level [33].

According to the International Fertilizer Industry Association (IFA), the moisture content of fertilizers granules must be less than 2% [34]. The obtained results show that the moisture content of the granules in this study varied within the range of 1.01% to 1.41%, which is more than the moisture content of industrially coated urea (0.3%).

The loose bulk density values of the granules are similar for the particles of both fractions, but they are not high when compared to the values of NPK mineral fertilizers with a loose bulk density of 900–1100 kg/m^3^ or a urea loose bulk density of 769 kg/m^3^ [35]. The lower loose bulk density of these fertilizers is most likely because the granules were produced using different methods to industrially produced fertilizers (in industry, urea is produced from a melt). A greater bulk density means that more material can fit in the same space and this is an advantage when looking to maximize the allotted volume of a container. Additionally, a higher density is also an advantage when spreading fertilizer with centrifugal spreaders, as heavier particles are spread more widely [36].

For mechanical spreading, it is important that particle size variations within a granular product are minimal. Because the SGN is the average of the particle sizes that make up the product, this parameter is commonly used in fertilizer specifications to describe particle size characteristics. It is very close to the SGN values of urea 280–320 [26]. Values vary in a small range, which reduces the risk of an uneven distribution or segregation, and can be blended with other similar products. SGN 200+ are typically used to fertilize landscape turf, golf, and other turfs of a standard cut [37].

Nitrogen leaching from water-soluble fertilizers and the resulting losses and water pollution are serious problems. If nitrogen existed in the soil in the nitrate or urea forms, significant leaching losses can occur with heavy rains, more so on coarse-textured soils. Some of this N may have leached deep enough into the root zone to be unavailable to the crop, at least early in the season. Continued precipitation or irrigation may leach this N out of the root zone entirely [38]. Most of the time, attempts are made to control the leaching process using a variety of inorganic-, bio-, or organic-coating films including urease activity and the inhibition of nitrification [39]. Coating technologies are quite complex and require additional equipment, so a simple way to slow down the solubility of fertilizers has been pursued in this work. Attempts were made to reduce nitrogen leaching from urea by mixing various binders directly into the raw material mixture prior to granulation.

Five leaching stages were performed in total, but the 4th and 5th stages did not yield any results. Our results correlate with [3] data that the highest nitrate leaching rate was observed in the first three days, and then the declining trend followed until the end of the incubation period. However, the rate and amount of nitrate leaching from different N fertilizers applied to soils at different times varied. There was a significant difference observed between fertilizers coated with nano-ZnO and normal urea-applied soil with respect to the total amount of NO_3_^−^ N leached during the study period.

Research shows that the main factor in slowing down the nitrogen release was binder network structures that increased the path length for water penetration [40]. Unfortunately, it must be stated and agreed that some of the nitrogen was lost during the leaching study due to ammonia and CO_2_ emissions [41].

## 4. Materials and Methods

### 4.1. Materials

The main raw materials needed to produce granular bioactive nitrogen fertilizers were crystalline (46.00% N) urea (U) [26] bought from “Reaxim” and green microalgae *Chlorella* sp. powder (ChSP) bought from “Buxtrade” (Buxtehude, Germany) (Buxtrade’s nutritional microalgae information: 11.50% fat, 2.34% saturated fatty acid, 4.59% polyunsaturated fatty acids, 17.30% carbohydrates, 0.32% sugar, 10.68% roughage, and 53.27% protein). Also, the binders such as water (W) or aqueous solutions of different substances: molasses (M)–waste from Marijampolė sugar factory (Lithuania); polyvinyl acetate dispersion (PVAD) from “Achema” (Jonava, Lithuania); carboxymethyl cellulose (CMC) from “Sercalia” (Spain), and potato starch (S) from AB “Roquette Amilina” (Panevėžys, Lithuania) were used. In order to study the influence of recycled materials, i.e., crushed non-marketable fraction (granules smaller than 2 mm and larger than 4 mm) on the granulation process, 20, 40, and 60% (*w*/*w*) recycled materials were used in the raw material mixture.

To determine the extent of nitrogen leaching, sand, washed with water and dried at 100 °C, was used. The sand was taken from Bartkūniškiai village, Kėdainiai district. The nitrogen concentration in the sand was determined before the experiment.

### 4.2. Methods

#### 4.2.1. Fertilizer Granulation

Microalgae-enriched urea fertilizers were granulated by the method of wet granulation (the liquid phase binder was sprayed onto a powdered mixture of dry raw materials) using a laboratory drum granulator and dryer (Figure 6). All raw materials (urea, algae, and one of the binders) were mixed into a homogeneous mixture and fed into the granulation drum. This equipment is an original scaled-down version of a commercially available drum NPK fertilizer granulator and was made by special order in the “Agrofertis” factory (Marijampolė, Lithuania). The granulation process used U, 10% (*w*/*w*) of microalgae ChSP, and W or other binder solutions (aqueous solutions) of the following concentration (*w*/*w*): PVAD (1, 3, 5, and 7%), M (2.5, 5, 7.5, and 10%), S (1, 3, and 5%), and CMC (1, 3, and 5%). The granulation time was 7 min, at a temperature of 50–60 °C, with a drum rotation speed of 26 rpm, and an inclination angle of 5°; the mass of dry raw material mixture used was 200 g. Granular fertilizers were dried in a drying oven Snol-1,6.2,5,1/9-IZ (“Elektrodelo”, Cheboksary, Russia) at 60 °C (accuracy ± 2 °C) until their moisture content was no higher than 2%. After that, granules were fractionated by separating the marketable fraction (granules with a size of 2.0–4.0 mm) and analysed. Granules smaller than 2 mm and larger than 4 mm were crushed with Waring laboratory blenders MX-7011S (“Gardco”, USA), mixed with raw materials of appropriate composition, and returned to granulation as recycled materials (Figure 6).

#### 4.2.2. Fractional Composition of Fertilizers

The test of the particle size distribution (fractional composition) of fertilizers was performed with braided sieves DIN-ISO 3310/1 (“Retsch GmbH Haan”, Hann, Germany) with a mesh size of <0.2, 0.5, 1.0, 2.0, 3.15, 4.0, and 5.0 mm. The sieving time was 5 min, and the frequency was 60 Hz. The particles were weighed on electronic scales WPS 210/C KERN ABJ (USA) with a balanced accuracy of ±0.001 g.

#### 4.2.3. Fertilizer Size Guide Number

The size guide number (SGN), which is the average particle diameter of the granules in millimeters (mm) multiplied by 100, is an important characteristic of a fertilizer. Thus, it was calculated to evaluate the efficiency of the granulation process [42].

#### 4.2.4. Crushing Strength of Fertilizers

The static crushing strength of granules was measured as the force which breaks the granule. It was determined using IPG-2 “UNIXIM s OZ” equipment (Ekaterinburg, Russia) with a compressive capacity of 5–200 N/gran. (accuracy ± 1.6%). At least 20 granules in each sample were tested and representative results are presented as the mathematical average of strength according to [43]. The obtained data were converted to MPa by evaluating the diameter of the granules.

#### 4.2.5. Hygroscopicity of Fertilizers

The hygroscopicity study of marketable fraction granules, i.e., granules with a diameter of 2.0–4.0 mm, was performed by placing the samples in a desiccator of two different environments: the vapour of a saturated NaNO_2_ solution and water vapour, to observe changes in their mass. The measurements were taken for 12 days. The temperature in the desiccator filled with water was 20.2–26.1 °C and the relative humidity was 99–100%, i.e., critical conditions for the handling and storage of fertilizers, whereas the temperature in the desiccator filled with saturated sodium nitrite solution was 20.6–27.2 °C with a humidity of 60–61%, i.e., ordinary conditions for the handling and storage of fertilizers. These tests were carried out in triplicate and the difference in hygroscopicity between the test results did not exceed 0.2% [44].

The hygroscopicity of all kinds of microalgae-enriched granular urea fertilizers is low in ordinary conditions for the handling and storage of fertilizers. The maximum change in mass of 7.4% was determined when the fertilizers were granulated using a 2.5% M solution. However, the fertilizers with the PVAD binder demonstrated low hygroscopicity, even in critical conditions [45].

#### 4.2.6. Moisture Content of Fertilizers

The moisture content of the microalgae-enriched urea granules was determined by measuring the mass of a sample before and after the water was removed by evaporation using an electronic moisture analyzer Kern & Sohn GmbH (Balingen, Germany), with an accuracy of 0.001 g. The tests were performed in triplicate.

#### 4.2.7. Bulk Density of Fertilizers

The fertilizer granule’s bulk density (or volume weight) provides information relative to the required size of packaging materials, storage houses, stock rooms, etc. The two usual bulk density measurements are loose bulk density, which generally refers to the material’s loose bulk density, and tapped density, which is the material’s bulk density after the container of material has been tapped until no further change occurs. Generally, the tapped bulk density is up to 10% greater than the loose bulk density and can be calculated. Loose bulk density was measured according to ISO 3944:1992 using a graded 100 cm^3^ cylinder and WPS210/C Kern ABJ balance.

#### 4.2.8. pH of Fertilizers

The pH values of 10% fertilizer aqueous solutions were measured with a pH meter from Hanna Instruments pH 211 (Leighton Buzzard, UK). The mass (10 g) of fertilizers was weighed on a WPS210/C Kern ABJ scale and dissolved in distilled water (90 g). Consequently, the pH of the solution was measured.

#### 4.2.9. Density of Binder Solutions

The density of binders at various concentrations was determined by the weight method using a hydrometer.

#### 4.2.10. Optical Microscopy

Optical microscopy images were taken with a Zeiss Discovery V12 Stereo microscope equipped with a PlanApoS 1.5× prime lens (“Zeiss”, Munich, Germany). The rate of magnification was ×22.5.

#### 4.2.11. Leaching of Nitrogen from Fertilizers

Nitrogen leaching studies were performed using a sand bed column with analysed particles placed in the middle. Leaching occurs most frequently in coarse-textured or sandy, well-drained soils, therefore, sand was used as a filling material [46]. All glass funnels had a glass filter and were of equal volume and uniform porosity (100 µm). The exact number of granules containing 0.5 g of nitrogen was added to the selected amount (40 g) of filler. An aliquot part of water (50 cm^3^) was poured into the funnel every 24 h and the leaked solution was collected in a beaker and analysed using the Kjeldahl method [44].

#### 4.2.12. Determination of Nitrogen Concentration

The concentration of both nitrogen forms (amide–NH_2_ and ammonium–NH_4_^+^) in the granulated fertilizers, sand, and leaked solutions were determined by the Kjeldahl method with the mineralizer Turbodog TUR/TVK and automatic Vapodest 45 s Gerhardt system (Konigswinter, Germany) according to DIN EN ISO 9001. This method had an accuracy of 0.5%. The tests were performed in triplicate and the difference in concentration between test results did not exceed 0.3% [44].

#### 4.2.13. Statistical Analysis

The measurements of the granulation of different mixtures, the density of different binders, and the analysis of granulated fertilizer properties (pH, hygroscopicity, bulk density, and humidity) were repeated 2–3 times, and the test of static strength was performed 20 times. The arithmetic mean of the determined values is presented in this study. Statistical standard deviation (SD) was also calculated. The results were calculated with 95% probability, with a significance level of *p* ≤ 0.05. The method of one-way analysis of variance (ANOVA) was used to evaluate the results.

## Figures and Tables

**Figure 1 plants-11-03362-f001:**
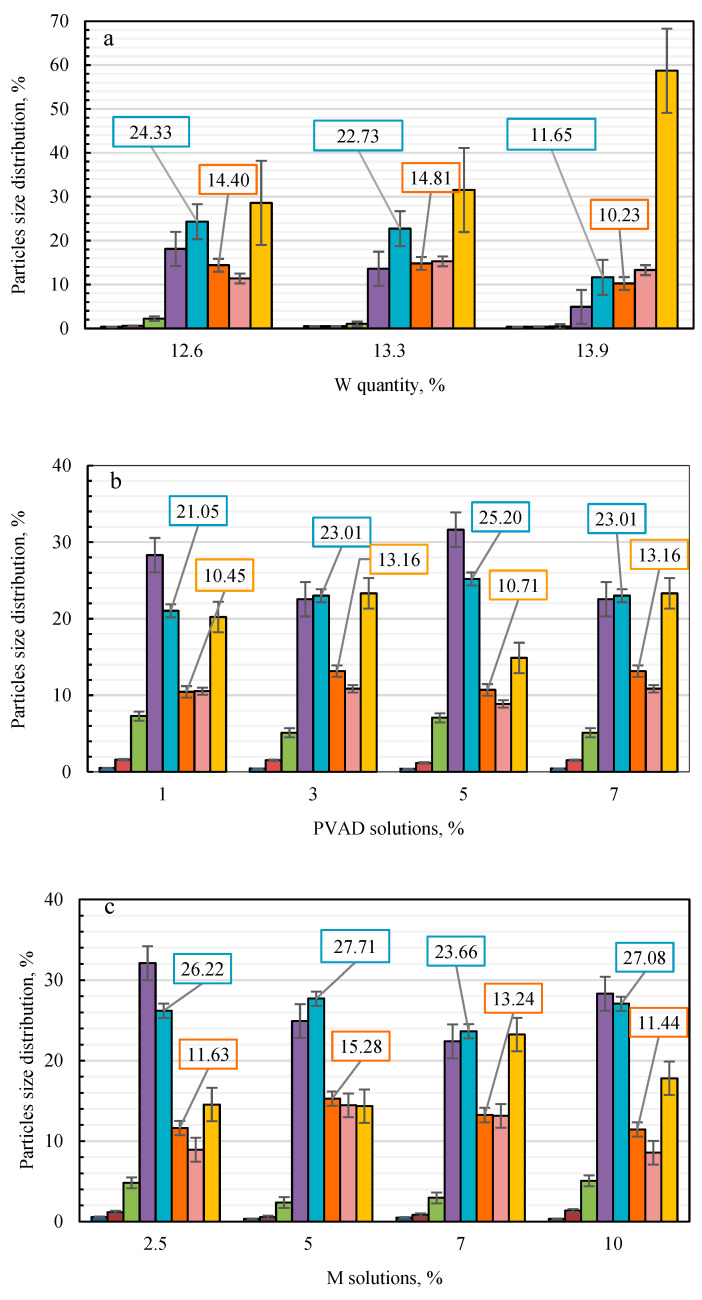
Distribution according to particle size (mm) using different binders: (**a**)—W, (**b**)—PVAD, (**c**)—M, (**d**)—S, and (**e**)—CMC.

**Figure 2 plants-11-03362-f002:**
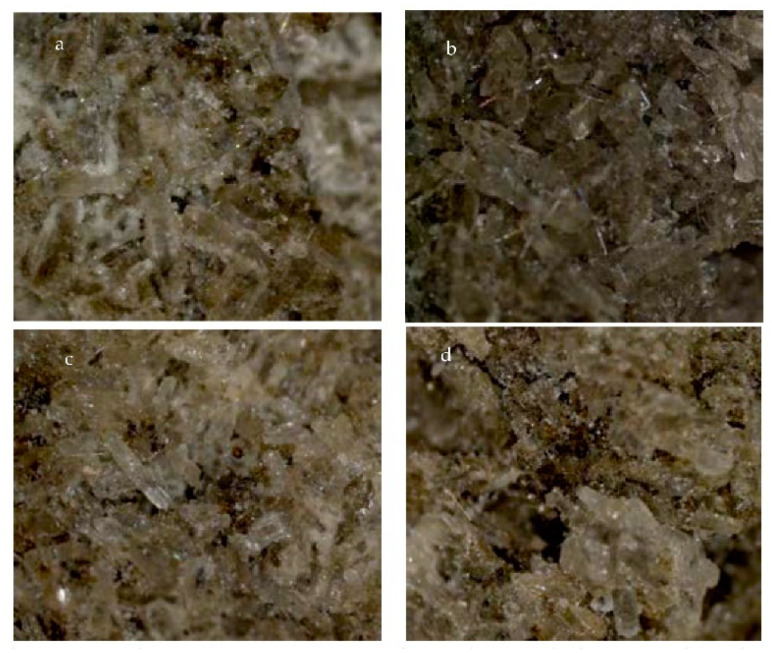
Optical microscope pictures (×22.5) of microalgae-enriched urea granules with different binders: (**a**)—5% PVAD, (**b**)—5% M, (**c**)—1% S, and (**d**)—1% CMC.

**Figure 3 plants-11-03362-f003:**
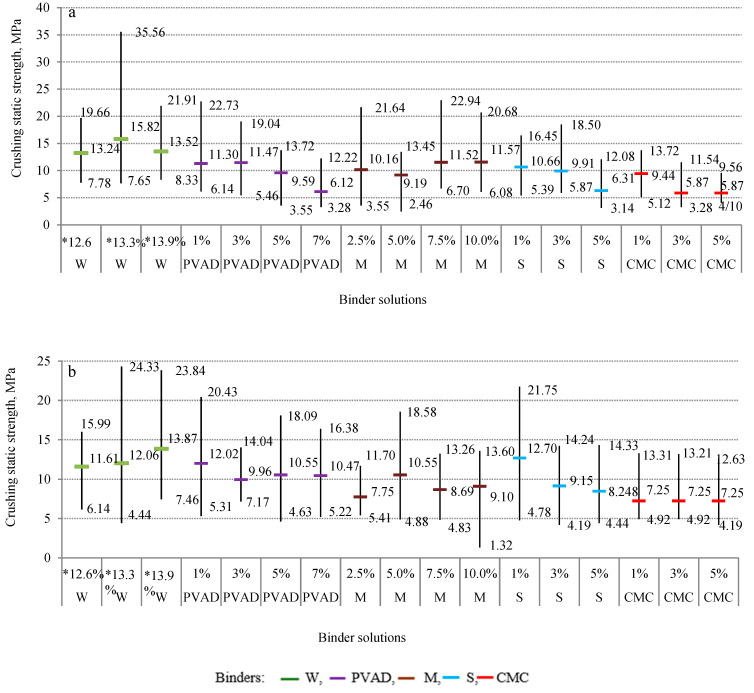
Static crushing strength of marketable fraction particles produced using different binders: (**a**)—2.0–3.15 mm, (**b**)—3.15–4.0 mm. * Amount of the water in the mixture of raw materials.

**Figure 4 plants-11-03362-f004:**
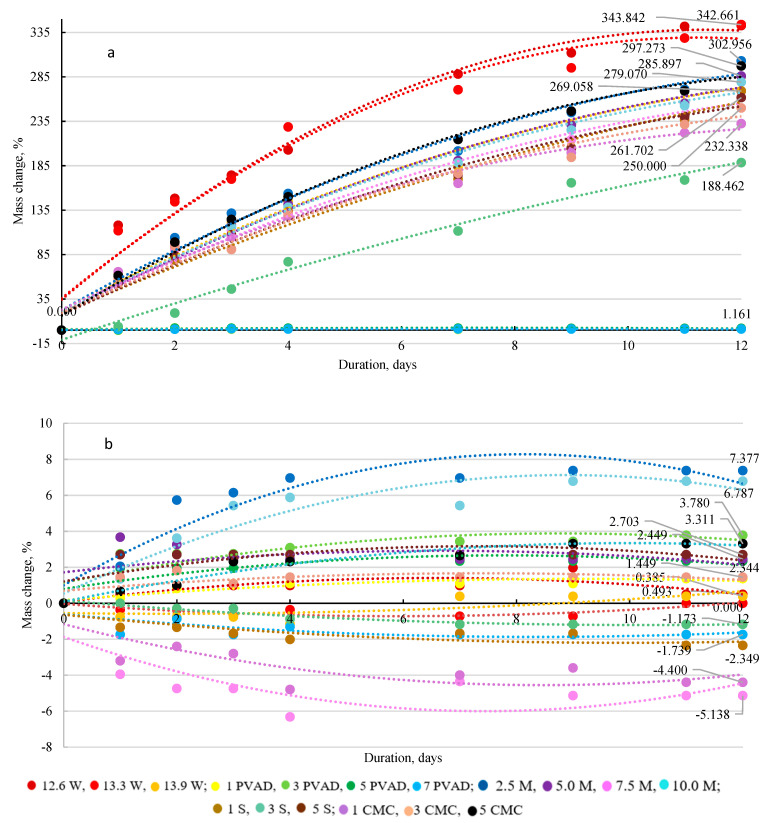
Fertilizer hygroscopicity in the environment: (**a**)—H_2_O and (**b**)—saturated NaNO_2_.

**Figure 5 plants-11-03362-f005:**
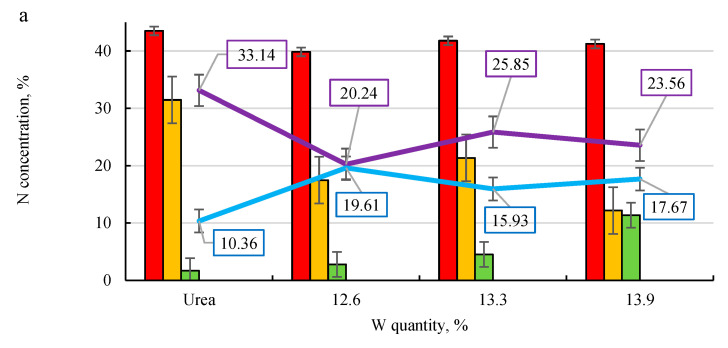
Nitrogen leaching from fertilizers granulated with different binders: (**a**)—W; (**b**)—PVAD; (**c**)—M; (**d**)—S; (**e**)—CMC.

**Figure 6 plants-11-03362-f006:**
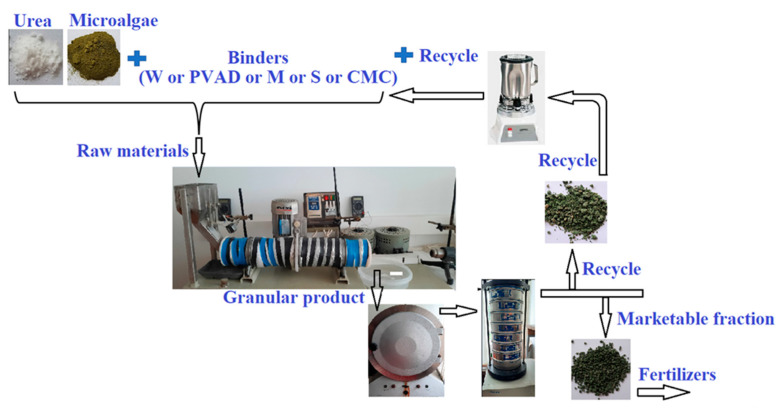
Drum granulator-dryer and material flow movement.

**Table 1 plants-11-03362-t001:** Amount of marketable fraction.

Binder	Binder Solution Concentration, %	Binder Solution Density, g/cm^3^	Amount of Binder, % (*w*/*w*)	Amount of Marketable Fraction, %
W	–	1.000 ± 0.000	12.6	38.72 ± 1.113
	–	1.000 ± 0.000	13.3	37.54 ± 3.662
	–	1.000 ± 0.000	13.9	21.89 ± 2.575
PVAD	1.0	0.999 ± 0.001	11.3	31.51 ± 3.503
	3.0	1.006 ± 0.002	11.3	36.17 ± 4.428
	5.0	1.021 ± 0.002	11.3	36.18 ± 2.787
	7.0	1.036 ± 0.001	11.3	35.91 ±4.529
M	2.5	1.012 ± 0.002	11.4	37.86 ± 2.391
	5.0	1.020 ± 0.004	11.4	43.01 ± 3.068
	7.5	1.042 ± 0.001	11.4	38.53 ± 1.744
	10.0	1.054 ± 0.002	11.4	36.91 ± 1.006
S	1.0	1.027 ± 0.004	11.0	32.34 ± 1.451
	3.0	1.031 ± 0.001	11.0	29.84 ± 2.386
	5.0	1.045 ± 0.001	11.0	26.88 ± 1.627
CMC	1.0	1.032 ± 0.002	11.1	40.25 ± 2.150
	3.0	1.042 ± 0.001	11.1	28.13 ± 2.617
	5.0	1.057 ± 0.001	11.1	26.68 ± 1.490

**Table 2 plants-11-03362-t002:** Physical properties of microalgae-enriched urea granules with different binders.

Binder	Concentration of Binder, %	pH of 10% Fertilizer Solution	Humidity of Fertilizers, %	Loose Bulk Density of Fertilizers, kg/m^3^		SGN
				2.0–3.15 mm	3.15–4.0 mm	
W		5.85 ± 0.000	1.23 ± 0.038	391.0 ± 2.484	386.0 ± 2.609	295
		6.05 ± 0.000	1.30 ± 0.028	401.0 ± 7.452	400.1 ± 7.825	297
		6.10 ± 0.000	1.22 ± 0.038	412.0 ± 2.484	393.7 ± 2.484	304
PVAD	1	6.00 ± 0.038	1.03 ± 0.038	355.9 ± 0.497	353.5 ± 1.120	291
	3	6.05 ± 0.014	1.02 ± 0.025	364.2 ± 6.713	370.5 ± 3.229	294
	5	6.10 ± 0.038	1.07 ± 0.063	363.7 ± 0.799	346.4 ± 0.872	287
	7	6.10 ± 0.014	1.05 ± 0.014	364.1 ± 7.137	351.2 ± 2.981	294
M	2.5	6.50 ± 0.029	1.32 ± 0.038	355.7 ± 1.368	350.2 ± 0.872	288
	5.0	6.40 ± 0.000	1.36 ±0.029	367.1 ± 0.872	357.7 ± 0.379	293
	7.5	6.60 ± 0.014	1.37 ± 0.038	372.1 ± 1.616	371.4 ± 1.120	293
	10.0	6.65 ± 0.087	1.41 ± 0.052	416.0 ± 0.872	387.2 ± 0.379	287
S	1	6.30 ± 0.014	1.32 ± 0.038	358.4 ± 1.616	347.0 ± 0.799	291
	3	6.35 ± 0.014	1.31 ± 0.025	356.3 ± 1.120	354.3 ± 1.225	297
	5	6.35 ± 0.130	1.29 ± 0.025	430.0 ± 1.120	418.1 ± 0.798	294
CMC	1	6.35 ± 0.014	1.22 ± 0.038	377.0 ± 0.379	338.5 ± 0.379	289
	3	6.40 ± 0.014	1.17 ± 0.049	341.2 ± 1.368	336.3 ± 1.225	290
	5	6.40 ± 0.025	1.21 ± 0.014	305.1 ± 2.609	331.8 ± 1.034	294

## Data Availability

Data sharing not applicable.

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
