# Peer review of "Effect of Various Binders on the Properties of Microalgae-Enriched Urea Granules"

_plants, 2022, doi:10.3390/plants11233362_

Round 1

Reviewer 1 Report

The tables in Figure 2, 4, 5, 6 are difficult to read unless printed in color, or even for a color-blind person. I would suggest using a combination of figure to distinguish the different theses. Although the standard deviation bar is present, there is a complete lack of statistical discussion.

Chapter and subchapter numbers in Materials and Methods are incorrect and do not follow the numbering of the manuscript.

Author Response

Dear Reviewer1,

Thank you for your comments. Please see the attachment.

Reviewer 2 Report

Hanna Klikocka                                                                                             Lublin, 20.11.2022

University of Life Sciences

Lublin, Poland

Review: Plants 2050816

Title:

 Effect of various binders on the properties of microalgae enriched urea granules

Authors: AustÄ—ja MikolaitienÄ— and Rasa Šlinkšiene

The aim of this study is to create microalgae enriched nitrogen fertilizers with different binders that inhibit nitrogen leaching from the soil. The results show that the highest quantity of the marketable fraction is 43.01±3.068% and it was obtained using urea, with 10% (w/w) microalgae additive, and 11.4% (w/w) of 5% concentration molasses solution. The granules of the fertilizer marketable fraction are similar in size because the average particle diameter of the granules (SGN) varies in a narrow range and falls within the interval from 287 to 304; this means that the average particle size is ~3 mm. When different binders were used, the aver[1]age static crushing strength of the granulated fertilizers was lower (approximately 6–12 MPa) than using water only (approximately 12–16 MPa), but it still fell into the required range. Additives of 25 PVAD solutions and of molasses solutions have been found to retain nitrogen in the sand.

The publication presented for evaluation is very interesting.

To improve your work, please check and correct:

1.      Figure 2: a – W, b – PVAD, c – M, d – S, e – CMC – add below explanations for the abbreviations used, e.g. S – starch.

2.      Table 1 – the text states that there were significant differences between the results in Table 1. This should be indicated by adding LSD or various letters. If there were no differences, please explain under Table 1.

3.      Line 408-412 – this is the reviewer's note, please remove it!

After improvement, I recommend publication for printing in the selected magazine.

Sincerely

Author Response

Dear Reviewer2,

Thank you for your comments. Please see the attachment.

Reviewer 3 Report

The theme of  article “Effect of various binders on the properties of microalgae en3 riched urea granules” is  interesting  and authors have tried to  compile in a good way, but the  presentation of article  and English  must be improved especially  the sentences  need to be reframed

I am mentioning herewith  some points:

Line-9-12 “Nitrogen is one of the main plant nutrients, most commonly used as the chemical substance urea. However, very large amounts of  nitrogen are lost, which is a serious environmental problem” check and reframe the sentence

Line 30- change the word “automatically”

Line 33-36-  reframe the sentence

Fig.4 delete the values from figure

Fig.5, symbol used in the footnote arranged in a sequence

Author Response

Dear Reviewer3,

Thank you for your comments. Please see the attachment.

Round 2

Reviewer 3 Report

Author have made significant changes and paper can be accepted in the present form